# TransINT: Embedding Implication Rules in Knowledge Graphs with Isomorphic Intersections of Linear Subspaces

**So Yeon Min**                                                                  symin95@alum.mit.edu
*MIT CSAIL,*
*32 Vassar Street, Cambridge, MA 02142 USA*

**Preethi Raghavan**                                                              praghav@us.ibm.com
*IBM Research,*
*75 Binney Street, Cambridge, MA 02142 USA*

**Peter Szolovits**                                                               psz@mit.edu
*MIT CSAIL,*
*32 Vassar Street, Cambridge, MA 02142 USA*

## Abstract

Knowledge Graphs (KG), composed of entities and relations, provide a structured representation of knowledge. For easy access to statistical approaches on relational data, multiple methods to embed a KG into $f(\text{KG}) \in \mathbb{R}^d$ have been introduced. We propose TransINT, a novel and interpretable KG embedding method that isomorphically preserves the implication ordering among relations in the embedding space. Given implication rules, TransINT maps sets of entities (tied by a relation) to continuous sets of vectors that are inclusion-ordered isomorphically to relation implications. With a novel parameter sharing scheme, TransINT enables automatic training on missing but implied facts without rule grounding. On two benchmark datasets, we outperform the best existing state-of-the-art rule integration embedding methods with significant margins in link prediction and triple classification. The angles between the continuous sets embedded by TransINT provide an interpretable way to mine semantic relatedness and implication rules among relations.

## 1. Introduction

Learning distributed vector representations of multi-relational knowledge is an active area of research [Bordes et al., 2013, Nickel et al., 2011, Kazemi and Poole, 2018a, Wang et al., 2014, Bordes et al., 2011]. These methods map components of a KG (entities and relations) to elements of $\mathbb{R}^d$ and capture statistical patterns, regarding vectors close in distance as representing similar concepts. One focus of current research is to bring logical rules to KG embeddings [Guo et al., 2016, Wang et al., 2015a, Wei et al., 2015]. While existing methods impose hard geometric constraints and embed asymmetric orderings of knowledge [Nickel and Kiela, 2017, Vendrov et al., 2015, Vilnis et al., 2018], many of them only embed hierarchy (unary *Is_a* relations), and cannot embed binary or n-ary relations in KG's. On the other hand, other methods that integrate binary and n-ary rules [Guo et al., 2016, Fatemi et al., 2018, Rocktäschel et al., 2015, Demeester et al., 2016] do not bring significant enough performance gains.

We propose TransINT, a new and extremely powerful KG embedding method that isomorphically preserves the implication ordering among relations in the embedding space. Given pre-defined implication rules, TransINT restricts entities tied by a relation to be

embedded to vectors in a particular region of $\mathbb{R}^d$ included isomorphically to the order of relation implication. For example, we map any entities tied by *is_father_of* to vectors in a region that is part of the region for *is_parent_of*; thus, we can automatically know that if John is a father of Tom, he is also his parent even if such a fact is missing in the KG. Such embeddings are constructed by sharing and rank-ordering the basis of the linear subspaces where the vectors are required to belong.

Mathematically, a relation can be viewed as sets of entities tied by a constraint [Stoll, 1979]. We take such a view on KG's, since it gives consistency and interpretability to model behavior. We show that angles between embedded relation sets can identify semantic patterns and implication rules - an extension of the line of thought as in word/ image embedding methods such as Mikolov et al. [2013], Frome et al. [2013] to relational embedding.

The main contributions of our work are: (1) A novel KG embedding such that implication rules in the original KG are guaranteed to unconditionally, not approximately, hold. (2) Our model suggests possibilities of learning semantic relatedness between groups of objects. (3) We significantly outperform state-of-the-art rule integration embedding methods, [Guo et al., 2016] and [Fatemi et al., 2018], on two benchmark datasets, FB122 and NELL Sport/ Location.

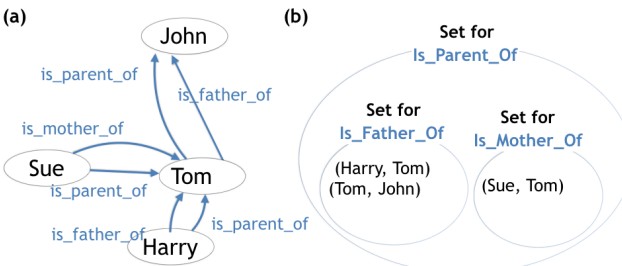

**Figure 1:** Two equivalent ways of expressing relations. (a): relations defined in a hypothetical KG. (b): relations defined in a set-theoretic perspective (Definition 1). Because *is_father_of* $\Rightarrow$ *is_parent_of*, the set for *is_father_of* is a subset of that for *is_parent_of* (Definition 2).

## 2. TransINT

In this section, we describe the intuition and justification of our method. We first define relation as sets, and revisit TransH [Wang et al., 2014] as mapping relations to sets in $\mathbb{R}^d$. Finally, we propose TransINT. We put $^*$ next to definitions and theorems we propose/ introduce. Otherwise, we use existing definitions and cite them.

### 2.1 Sets as Relations

We define relations as sets and implication as inclusion of sets, as in set-theoretic logic.
**Definition** *(Relation Set):* Let $r_i$ be a binary relation and $x, y$ entities. Then, a set $\mathbf{R_i}$ such that $r_i(x, y)$ if and only if $(x, y) \in \mathbf{R_i}$ always exists [Stoll, 1979]. We call $\mathbf{R_i}$ the **relation set** of $r_i$.

For example, consider the distinct relations in Figure 1a, and their corresponding sets in Figure 1b; *Is_Father_Of(Tom, Harry)* is equivalent to *(Tom, Harry)* $\in \mathbf{R}_{\text{Is\_Father\_Of}}$.

***Definition*** *(Logical Implication):* For two relations, $r_1$ implies $r_2$ (or $r_1 \Rightarrow r_2$) iff $\forall x, y$,

$$(x, y) \in \mathbf{R_1} \Rightarrow (x, y) \in \mathbf{R_2} \quad \text{or equivalently,} \quad \mathbf{R_1} \subset \mathbf{R_2}.[\text{Stoll, 1979}]$$

For example, *Is_Father_Of* $\Rightarrow$ *Is_Parent_Of.* (In Figure 1b, $\mathbf{R_{Is\_Father\_Of}} \subset \mathbf{R_{Is\_Parent\_Of}}$).

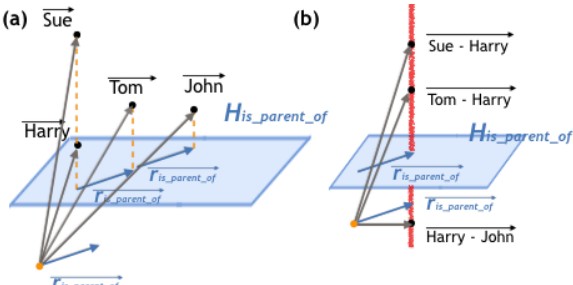

**Figure 2:** Two perspectives of viweing TransH in $\mathbb{R}^3$; order of operations can be flipped. (The orange dot is the origin, to emphasize that translated vectors are equivalent.) (a): projection first, then difference - first projecting $\vec{h}$ and $\vec{t}$ onto $H_{is\_parent\_of}$, and then requiring $\vec{h_\perp} + \vec{r_j} \approx \vec{t_\perp}$ (b): difference first, then projection - first subtracting $\vec{h}$ from $\vec{t}$, and then projecting the difference $(\overrightarrow{t - h})$ to $H_{is\_parent\_of}$ and requiring $(\overrightarrow{t - h})_\perp \approx r_j$. All $(\overrightarrow{t - h})_\perp$ belong to the red line, which is unique because it is when $\overrightarrow{r_{is\_parent\_of}}$ is translated to the origin.

## 2.2 Background: TransH

Given a fact triple $(h, r_j, t)$ in a KG (i.e. *(Harry, is_father_of, Tom)*), TransH maps each entity to a vector, and each relation $r_j$ to a relation-specific hyperplane $H_j$ and a fixed vector $\vec{r_j}$ on $H_j$ (Figure 2a). For each fact triple $(h, r_j, t)$, TransH wants

$$\vec{h_\perp} + \vec{r_j} \approx \vec{t_\perp} \cdots \cdots \quad \text{(Eq. 1)}$$

where $\vec{h_\perp}, \vec{t_\perp}$ are projections on $\vec{h}, \vec{t}$ onto $H_j$ (Figure 2a).

**Revisiting TransH**   We interpret TransH in a novel perspective. An equivalent way to put Eq.1 is to change the order of subtraction and projection (Figure 2b):

$$\text{Projection of } (\overrightarrow{t - h}) \text{ onto } H_j \approx \vec{r_j}.$$

This means that all entity vectors $(\vec{h}, \vec{t})$ such that their difference $\overrightarrow{t - h}$ belongs to the red line are considered to be tied by relation $r_j$ (Figure 2b); $\mathbf{R_j} \approx$ the red line, which is the set of all vectors whose projection onto $H_j$ is the fixed vector $\vec{r_j}$. Thus, upon a deeper look, **TransH actually embeds a relation set in KG (figure 1b) to a particular set in** $\mathbb{R}^d$. We call such sets **relation space** for now; in other words, a **relation space** of some relation $r_i$ is the space where each $(h, r_i, t)$'s $\overrightarrow{t - h}$ can exist. We formally visit it later in Section **3.1**. Thus, in TransH,

$$r_i(x, y) \equiv (x, y) \in \mathbf{R_i} \qquad \text{(\textbf{relation in KG})}$$
$$\cong \overrightarrow{y - x} \in \text{relation space of } r_i \quad \text{(\textbf{relation in } $\mathbb{R}^\mathbf{d}$)}$$

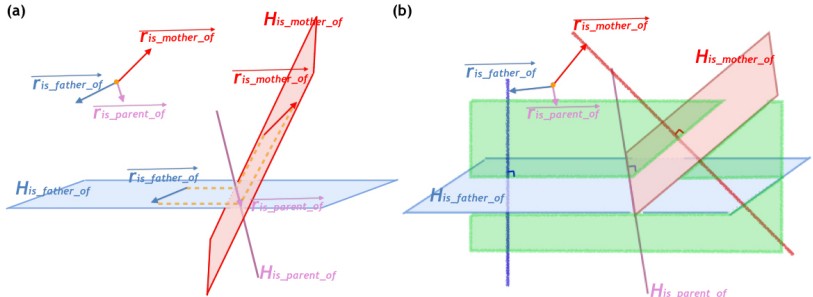

**Figure 3:** Two perspectives of viewing TransINT. (a): TransINT as TransH with additional constraints - by intersecting $H$'s and projecting $\vec{r}$'s. The dotted orange lines are the projection constraint. (b): TransINT as mapping of sets (relations in KG's) into linear subspaces (viewing TransINT in the relation space (Figure 2b)). The blue line, red line, and the green plane is respectively *is_father_of, is_mother_of* and *is_parent_of*'s relation space - where $\vec{t-h}$'s of $h, t$ tied by these relations can exist. The blue and the red line lie on the green plane - *is_parent_of*'s relation space includes the other two's.

### 2.3 TransINT

We propose TransINT, which, given pre-defined implication rules, guarantees isomorphic ordering of relations in the embedding space. Like TransH, TransINT embeds a relation $r_j$ to a (subspace, vector) pair $(H_j, \vec{r_j})$. However, TransINT modifies the relation embeddings $(H_j, \vec{r_j})$ so that the relation spaces (i.e. red line of Figure 2b) are ordered by implication; we do so by intersecting the $H_j$'s and projecting the $\vec{r_j}$'s (Figure 3a). We explain with familial relations as a running example.

**Intersecting the $H_j$'s**   TransINT assigns distinct hyperplanes $H_{is\_father\_of}$ and $H_{is\_mother\_of}$ to *is_father_of* and *is_mother_of*. However, because *is_parent_of* is implied by the aforementioned relations, we assign

$$H_{is\_parent\_of} = H_{is\_father\_of} \cap H_{is\_mother\_of}.$$

TrainsINT's $H_{is\_parent\_of}$ is not a hyperplane but a linear subspace of rank 2 (Figure 3a), unlike in TransH where all $H_j$'s are hyperplanes (whose ranks are 1).

**Projecting the $\vec{r_j}$'s**   TransINT constrains the $\vec{r_j}$'s with projections (Figure 3a's dotted orange lines). First, $\overrightarrow{r_{is\_father\_of}}$ and $\overrightarrow{r_{is\_mother\_of}}$ are required to have the same projection onto $H_{is\_parent\_of}$. Second, $\overrightarrow{r_{is\_parent\_of}}$ is that same projection onto $H_{is\_parent\_of}$.

We connect the two above constraints to ordering relation spaces. Figure 3b graphically illustrates that *is_parent_of*'s relation space (green hyperplane) includes those of *is_father_of* (blue line) and *is_mother_of* (red line). More generally, TransINT requires that

For distinct relations $r_i, r_j$, require the following if and only if $r_i \Rightarrow r_j$:
**Intersection Constraint:** $H_j = H_i \cap H_j$.
**Projection Constraint:** Projection of $\vec{r_1}$ to $H_j$ is $\vec{r_j}$.
where $\vec{H_i}, \vec{H_j}$ and $\vec{r_i}, \vec{r_j}$ are distinct.

We prove that these two constraints guarantee that an ordering isomorphic to implication holds in the embedding space: $(r_i \Rightarrow r_j)$ iff ($r_i$'s rel. space $\subset r_j$'s rel. space)

or equivalently, $(R_i \subset R_j)$ iff ($r_i$'s rel. space $\subset r_j$'s rel. space).

## 3. TransINT's Isomorphic Guarantee

In this section, we formally state TransINT's isomorphic guarantee. We denote all $d \times d$ matrices with capital letters (e.g. $A$) and vectors with arrows on top (e.g. $\vec{b}$).

### 3.1 Projection and Relation Space

In $\mathbb{R}^d$, there is a bijection between each linear subspace $H_i$ and a projection matrix $P_i$; $\forall \vec{x} \in \mathbb{R}^d, P_i x \in H_i$ [Strang, 2006]. A random point $\vec{a} \in \mathbb{R}^d$ is projected onto $H_i$ iff multiplied by $P_i$; i.e. $P_i a = \vec{b} \in H_i$. In the rest of the paper, we denote $P$ (or $P_i$) as the projection matrix onto a linear subspace $H$ (or $H_i$). Now, we formally define a general concept that subsumes relation space (Figure 3b).

**Definition*** $(Sol(P, \vec{k}))$ : Let $H$ be a linear subspace and $P$ its projection matrix. Then, given $\vec{k}$ on $H$, the set of vectors that become $\vec{k}$ when projected on to $H$, or the solution space of $P\vec{x} = \vec{k}$, is denoted as $\mathbf{Sol(P}, \vec{k})$.

With this definition, relation space (Figure 3b) is $(Sol(P_i, \vec{r_i}))$, where $P_i$ is the projection matrix of $H_i$ (subspace for relation $r_i$); it is the set of points $\overrightarrow{t-h}$ such that $P_i(\overrightarrow{t-h}) = \vec{r_i}$.

### 3.2 Isomorphic Guarantees

**Main Theorem 1** *(Isomorphism):* Let $\{(H_i, \vec{r_i})\}_n$ be the (subspace, vector) embeddings assigned to relations $\{\mathbf{R_i}\}_n$ by the *Intersection Constraint* and *the Projection Constraint*; $P_i$ the projection matrix of $H_i$. Then, $(\{Sol(P_i, \vec{r_i})\}_n, \subset)$ is isomorphic to $(\{\mathbf{R_i}\}_n, \subset)$.

In actual optimization, TransINT requires something less strict than $P_i(\overrightarrow{t-h}) = \vec{r_i}$:

$$P_i(\overrightarrow{t-h}) - \vec{r_i} \approx \vec{0} \equiv ||P_i(\overrightarrow{t-h} - \vec{r_i})||_2 < \epsilon,$$

for some non-negative and small $\epsilon$. This bounds $\overrightarrow{t-h} - \vec{r_i}$ to regions with thickness $2\epsilon$, centered around $Sol(P_i, \vec{r_i})$ (Figure 4). We prove that isomorphism still holds with this weaker requirement.

**Definition*** $(Sol_\epsilon(P, k))$ : Given any $P$, the solution space of $||P\vec{x} - \vec{k}||_2 < \epsilon$ (where $\epsilon \geq 0$) is denoted as $\mathbf{Sol_\epsilon(P}, \vec{k})$.

**Main Theorem 2** *(Margin-aware Isomorphism):* $\forall \epsilon \geq 0, (\{Sol_\epsilon(P_i, \vec{r_i})\}_n, \subset)$ is isomorphic to $(\{\mathbf{R_i}\}_n, \subset)$.

## 4. Initialization and Training

The intersection and projection constraints can be imposed with parameter sharing.

### 4.1 Parameter Sharing Initializaion

From initialization, we bind parameters so that they satisfy the two constraints. For each entity $e_j$, we assign a $d$-dimensional vector $\vec{e_j}$. To each $\mathbf{R_i}$, we assign $(H_i, \vec{r_i})$ (or $(A_i, \vec{r_i})$) with parameter sharing. Please see Appendix B on definitions of *head/ parent/ child relations*. We first construct the $H$'s.

**Intersection constraint** Each subspace $H$ can be uniquely defined by its orthogonal subspace. We define the orthogonal subspace of the $H$'s top-down. To every *head relation*

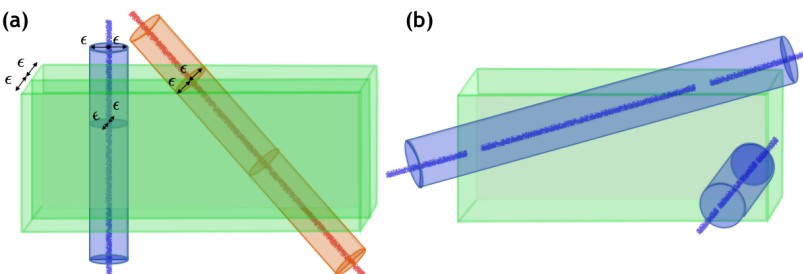

**Figure 4:** Fig. 3(b)'s relation spaces when $P_i(\overrightarrow{t-h}) - \vec{r_i} \approx \vec{0} \equiv ||P_i(\overrightarrow{t-h} - \vec{r_i})||_2 < \epsilon$ is required. (a): Each relation space now becomes regions with thickness $\epsilon$, centered around figure 3(b)'s relation space. (b): Relationship of the angle and area of overlap between two relation spaces. With respect to the green region, the nearly perpendicular cylinder overlaps much less with it than the other cylinder with much closer angle.

$\mathbf{R}_h$, assign a $d$-dimensional vector $\vec{a_h}$ as an orthogonal subspace for $H_{R_h}$, making $H_{R_h}$ a hyperplane. Then, to each $\mathbf{R}_i$ that is not a head, additionally assign a new $d$-dimensional vector $\vec{a_i}$ linearly independent to the bases of all of its parents. Then, $\mathbf{R}_i$'s basis of the orthogonal subspace for $H_{R_i}$ becomes $[\vec{a_h}, ..., \vec{a_p}, \vec{a_i}]$ where $\vec{a_h}, ..., \vec{a_p}$ are the vectors assigned to $\mathbf{R}_i$'s parent relations. Projection matrices can be uniquely constructed given the bases $[\vec{a_h}, ..., \vec{a_p}, \vec{a_i}]$ [Strang, 2006]. Now, we initialize the $\vec{r_i}$'s.

**Projection Constraint**  To the head relation $\mathbf{R}_h$, pick any random $x_h \in \mathbb{R}^d$ and assign $\vec{r_h} = P_h x$. To each non-head $\mathbf{R}_i$ whose parent is $\mathbf{R}_p$, assign $\vec{r_i} = \vec{r_p} + (I - P_p)(P_i)x_i$ for some random $x_i$. This results in

$$P_p\vec{r_i} = P_p\vec{r_p} + P_p(I - P_p)(P_i)\vec{x_i} = \vec{r_p} + \vec{0} = \vec{r_p}$$

for any parent, child pair.

**Parameters to be trained**  Such initialization leaves the following parameters given a KG with entities $e_j$'s and relations $r_i$'s: (1) a $d$-dimensional vector $(\vec{a_h})$ for the head relation, (2) a $d$-dimensional vector $(\vec{a_i})$ for each non-head relation, (3) a $d$-dimensional vector $\vec{x_i}$ for each head and non-head relation, (4) a $d$-dimensional vector $\vec{e_j}$ for each entity $e_j$. TransH and TransINT both assign two $d$-dimensional vectors for each relation and one $d$-dimensional vector for each entity; thus, TransINT has the same number of parameters as TransH.

### 4.2 Training

We construct negative examples (wrong fact triplets) and train with a margin-based loss, following the same protocols as in TransE and TransH.

**Training Objective**  We adopt the same loss function as in TransH. For each fact triplet $(h, r_i, t)$, we define the score function

$$f(h, r_i, t) = ||P_i(\overrightarrow{t-h}) - \vec{r_i}||_2$$

and train a margin-based loss $L$:

$$L = \sum_{(h, r_i, t) \in G} max(0, f(h, r_i, t)^2 + \gamma - f(h', r_i', t')^2).$$

where $G$ is the set of all triples in the KG and $(h', r'_i, t')$ is a negative triple made from corrupting $(h, r_i, t)$. We minimize this objective with stochastic gradient descent.

**Automatic Grounding of Positive Triples**   Without any special treatment, our initialization guarantees that training for a particular $(h, r_i, t)$ also automatically executes training with $(h, r_p, t)$ for any $r_i \Rightarrow r_p$, at all times. For example, by traversing *(Tom, is_father_of, Harry)* in the KG, the model automatically also traverses *(Tom, is_parent_of, Harry)*, *(Tom, is_family_of, Harry)*, even if they are missing in the KG. This is because $P_p P_i = P_p$ with the given initialization (section 4.1.1) and thus,

$$f(h, r_p, t) = ||P_p(\overrightarrow{t - h}) - \overrightarrow{r_p}||_2^2 = ||P_p(P_i((\overrightarrow{t - h}) - \overrightarrow{r_i}))||_2^2$$
$$\leq ||(P_p + (I - P_p))P_i((\overrightarrow{t - h}) - \overrightarrow{r_i}))||_2^2 = ||(P_i((\overrightarrow{t - h}) - \overrightarrow{r_i}))||_2^2 = f(h, r_i, t)$$

In other words, training $f(h, r_i, t)$ towards less than $\epsilon$ automatically guarantees training $f(h, r_p, t)$ towards less than $\epsilon$. This eliminates the need to manually create missing triples that are true by implication rule.

## 5. Experiments

We evaluate TransINT on two standard benchmark datasets - Freebase 122 [Bordes et al., 2013] and NELL sport/ location [Wang et al., 2015b] and compare against respectively KALE [Guo et al., 2016] and SimplE+ [Fatemi et al., 2018], state-of-the-art methods that integrate rules to KG embeddings, respectively in the trans- and bilinear family. We perform link prediction and triple classification tasks on Freebase 122, and link prediction only on NELL sport/ location (because SimplE+ only reported performance on link prediction). All codes for experiments were implemented in PyTorch [Paszke et al., 2019].[1]

### 5.1 Link Prediction on Freebase 122 and NELL Sport/ Location

We compare link prediction results with KALE on Freebase 122 (FB122) and with SimplE+ on NELL Sport/ Location. The task is to predict the gold entity given a fact triple with missing head or tail - if $(h, r, t)$ is a fact triple in the test set, predict $h$ given $(r, t)$ or predict $t$ given $(h, r)$. We follow TransE, KALE, and SimplE+'s protocol. For each test triple $(h, r, t)$, we rank the similarity score $f(e, r, t)$ when $h$ is replaced with $e$ for every entity $e$ in the KG, and identify the rank of the gold head entity $h$; we do the same for the tail entity $t$. Aggregated over all test triples, we report for FB 122: (i) the mean reciprocal rank (**MRR**), (ii) the median of the ranks (**MED**), and (iii) the proportion of ranks no larger than $n$ (**HITS@N**) which are the same metrics reported by KALE. For NELL Sport/ Location, we follow the protocol of SimplE+ and do not report MED. A lower MED, and a higher MRR and Hits HITS@N are better.

TransH, KALE, and SimplE+ adopt a "filtered" setting that addresses when entities that are correct, albeit not gold, are ranked before the gold entity. For example, if the gold entity is *(Tom, is_parent_of, John)* and we rank every entity $e$ for being the head of *(?, is_parent_of, John)*, it is possible that *Sue*, *John*'s mother, gets ranked before *Tom*. To avoid

---

1. Repository for all of our code: https://github.com/SoyeonTiffanyMin/TransINT

**Table 1:** Results for Link Prediction on FB122. *: For KALE, we report the best performance by any of KALE-PRE, KALE-Joint, KALE-TRIP (3 variants of KALE proposed by Guo et al. [2016]).

| | Raw | | | | | Filtered | | | | |
|---|---|---|---|---|---|---|---|---|---|---|
| | MRR | MED | Hits N% | | | MRR | MED | Hits N% | | |
| | | | 3 | 5 | 10 | | | 3 | 5 | 10 |
| **TransE** | 0.262 | 10.0 | 33.6 | 42.5 | 50.0 | 0.480 | 2.0 | 58.9 | 64.2 | 70.2 |
| **TransH** | 0.249 | 12.0 | 31.9 | 40.7 | 48.6 | 0.460 | 3.0 | 53.7 | 59.1 | 66.0 |
| **TransR** | 0.261 | 15.0 | 28.9 | 37.4 | 45.9 | 0.523 | 2.0 | 59.9 | 65.2 | 71.8 |
| **KALE**$^*$ | 0.294 | 9.0 | 36.9 | 44.8 | 51.9 | 0.523 | 2.0 | 61.7 | 66.4 | 72.8 |
| **TransINT**$^G$ | **0.339** | **6.0** | **40.1** | **49.1** | **54.6** | **0.655** | **1.0** | **70.4** | **75.1** | **78.7** |
| **TransINT**$^{NG}$ | 0.323 | 8.0 | 38.3 | 46.6 | 53.8 | 0.620 | 1.0 | 70.1 | 74.1 | 78.3 |

this, the "filtered setting" ignores corrupted triplets that exist in the KG when counting the rank of the gold entity. (The setting without this is called the "raw setting").

TransINT's hyperparameters are: learning rate ($\eta$), margin ($\gamma$), embedding dimension ($d$), and learning rate decay ($\alpha$), applied every 10 epochs to the learning rate. We find optimal configurations among the following candidates: $\eta \in \{0.003, 0.005, 0.01\}, \gamma \in \{1, 2, 5, 10\}, d \in \{50, 100\}, \alpha, \in \{1.0, 0.98, 0.95\}$; we grid-search over each possible ($\eta$, $\gamma$, $d$, $\alpha/0$. We create 100 mini-batches of the training set (following the protocol of KALE) and train for a maximum of 1000 epochs with early stopping based on the best median rank. Furthermore, we try training with and without normalizing each of entity vectors, relation vectors, and relation subspace bases after every batch of training.

### 5.1.1 EXPERIMENT ON FREEBASE 122

We compare our performance with that of KALE and previous methods (TransE, TransH, TransR) that were compared against it, using the same dataset (FB122). FB122 is a subset of FB15K [Bordes et al., 2013] accompanied by 47 implication and transitive rules; it consists of 122 Freebase relations on "people", "location", and "sports" topics. Out of the 47 rules in FB122, 9 are transitive rules (e.g. `person/nationality(x,y)` ∧ `country/official_language(y,z)` ⇒ `person/languages(x,z)`) to be used for KALE. However, since TransINT only deals with implication rules, we do not take advantage of them, unlike KALE.

We also put us on some intentional disadvantages against KALE to assess TransINT's robustness to absence of negative example grounding. In constructing negative examples for the margin-based loss $L$, KALE both uses rules (by grounding) and their own scoring scheme to avoid false negatives. While grounding with FB122 is not a burdensome task, it known to be very inefficient and difficult for extremely large datasets [Ding et al., 2018]. Thus, it is a great advantage for a KG model to perform well without grounding of training/ test data. We evaluate TransINT on two settings - with and without rule grounding. We call them respectively TransINT$^G$ (grounding), TransINT$^{NG}$ (no grounding).

We report link prediction results in Table 1; since we use the same train/ test/ validation sets, we directly copy from Guo et al. [2016] for baselines. While the *filtered* setting gives better performance (as expected), the trend is generally similar between *raw* and *filtered*. TransINT outperforms all other models by large margins in all metrics, even without

**Table 2:** Results for Link Prediction on NELL sport/ location.

| | Sport | | | | | Location | | | | |
|---|---|---|---|---|---|---|---|---|---|---|
| | MRR | | Hits N% | | | MRR | | Hits N% | | |
| | Filtered | Raw | 1 | 3 | 10 | Filtered | Raw | 1 | 3 | 10 |
| **Logical Inference** | - | - | 28.8 | - | - | - | - | 27.0 | - | - |
| **SimplE** | 0.230 | 0.174 | 18.4 | 23.4 | 32.4 | 0.190 | 0.189 | 13.0 | 21.0 | 31.5 |
| **SimplE+** | 0.404 | 0.337 | 33.9 | 44.0 | 50.8 | 0.440 | 0.434 | 43.0 | 44.0 | 45.0 |
| **TransINT$^G$** | **0.450** | 0.361 | **37.6** | **50.2** | **56.2** | **0.550** | **0.535** | **51.2** | **56.8** | **61.1** |
| **TransINT$^{NG}$** | 0.431 | **0.362** | 36.7 | 48.7 | 52.1 | 0.536 | 0.534 | 51.1 | 53.3 | 59.0 |

grounding; especially in the *filtered* setting, the **Hits@N** gap between TransINT$^G$ and KALE is around 4∼6 times that between KALE and the best Trans Baseline (TransR).

Also, while TransINT$^G$ performs higher than TransINT$^{NG}$ in all settings/metrics, the gap between them is much smaller than the that between TransINT$^{NG}$ and KALE, showing that TransINT robustly brings state-of-the-art performance even without grounding. The results suggest two possibilities in a more general sense. First, the emphasis of true positives could be as important as/ more important than avoiding false negatives. Even without manual grounding, TransINT$^{NG}$ has automatic grounding of positive training instances enabled (Section 4.1.1.) due to model properties, and this could be one of its success factors. Second, hard constraint on parameter structures can bring performance boost uncomparable to that by regularization or joint learning, which are softer constraints.

### 5.1.2 EXPERIMENT ON NELL SPORT/ LOCATION

We compare TransINT against SimplE+, a state-of-the-art method that outperforms ComplEx [Trouillon et al., 2016] and SimplE [Kazemi and Poole, 2018b], on NELL (Sport/ Location) for link prediction. NELL Sport/ Location is a subset of NELL [Mitchell et al., 2015] accompanied by implication rules - a complete list of them is available in Appendix C. Since we use the same train/ test/ validation sets, we directly copy from Fatemi et al. [2018] for baselines (Logical Inference, SimplE, SimplE+). The results are shown in Table 2. Again, TransINT$^G$ and TransinT$^{NG}$ significantly outperform other methods in all metrics. The general trends are similar to the results for FB 122; again, the performance gap between TransINT$^G$ and TransINT$^{NG}$ is much smaller than that between TransINT$^{NG}$ and SimplE+.

### 5.2 Triple Classification on Freebase 122

The task is to classify whether an unobserved instance $(h, r, t)$ is correct or not, where the test set consists of positive and negative instances. We use the same protocol and test set provided by KALE; for each test instance, we evaluate its similarity score $f(h, r, t)$ and classify it as "correct" if $f(h, r, t)$ is below a certain threshold ($\sigma$), a hyperparameter to be additionally tuned for this task. We report on mean average precision (MAP), the mean of classification precision over all distinct relations ($r$'s) of the test instances. We use the same experiment settings/ training details as in Link Prediction other than additionally finding optimal $\sigma$. Triple Classification results are shown in Table 3. Again, TransINT$^G$ and TransINT$^{NG}$ both significantly outperform all other baselines. We also separately analyze

**Table 3:** Results for Triple Classification on FB122, in Mean Average Precision (MAP).

| TransE | TransH | TransR | KALE* | TransINT$^G$ | TransINT$^{NG}$ |
|--------|--------|--------|-------|--------------|------------------|
| 0.634 | 0.641 | 0.619 | 0.677 | **0.781** (0.839/ 0.752) | **0.743** (0.709/ 0.761) |

**Table 4:** Examples of relations' angles and *imb* with respect to `/people/person/place_of_birth`

|  |  | Relation | Anlge | *imb* |
|---|---|---|---|---|
| Not Disjoint | Relatedness | `/people/person/nationality` | 22.7 | 1.18 |
|  | Implication | `/people/person/place_lived/location`* | 46.7 | 3.77 |
| Disjoint |  | `/people/cause_of_death/people` | 76.6 | n/a |
|  |  | `/sports/sports_team/colors` | 83.5 | n/a |

MAP for relations that are/ are not affected by the implication rules (those that appear/ do not appear in the rules), shown in parentheses of Table 3 with the order of (influenced relations/ uninfluenced relations). We can see that both TransINT's have MAP higher than the overall MAP of KALE, even when the TransINT's have the penalty of being evaluated only on uninfluenced relations; this shows that TransINT generates better embeddings even for those not affected by rules. Furthermore, we comment on the role of negative example grounding; we can see that grounding does not help performance on unaffected relations (i.e. 0.752 vs 0.761), but greatly boosts performance on those affected by rules (0.839 vs 0.709). While TransINT does not necessitate negative example grounding, it does improve the quality of embeddings for those affected by rules.

## 6. Semantics Mining with Overlap Between Embedded Regions

Traditional embedding methods that map an object (i.e. words, images) to a singleton vector learn soft tendencies between embedded vectors with cosine similarity, or angular distance between two embddings. TransINT extends such a line of thought to semantic relatedness between groups of objects, with angles between relation spaces. In Fig. 4b, one can observe that the closer the angle between two embedded regions, the larger the overlap in area. For entities $h$ and $t$ to be tied by both relations $r_1, r_2$, $\overrightarrow{t - h}$ has to belong to the intersection of their relation spaces. Thus, we hypothesize the following over any two relations $r_1, r_2$ that are not explicitly tied by the pre-determined rules:

Let $V_1$ be the set of $\overrightarrow{t - h}$'s in $r_1$'s relation space (denoted as $Rel_1$) and $V_2$ that of $r_2$'s.

(1) Angle between $Rel_1$ and $Rel_2$ represents semantic "disjointness" of $r_1, r_2$; the more disjoint two relations, the closer their angle to $90°$.

When the angle between $Rel_1$ and $Rel_2$ is small,

(2) if majority of $V_1$ belongs to the overlap of $V_1$ and $V_2$ but not vice versa, $r_1$ implies $r_2$.

(3) if majority of $V_1$ and $V_2$ both belong to their overlap, $r_1$ and $r_2$ are semantically related.

(2) and (3) consider the imbalance of membership in overlapped regions. Exact calculation of this involves specifying an appropriate $\epsilon$ (Fig. 3). As a proxy for deciding whether an element of $V_1$ (denote $v_1$) belongs in the overlapped region, we can consider the distance

between $v_1$ and its projection to $Rel_2$; the further away $v_1$ is from the overlap, the larger the projected distance. Call the mean of such distances from $V_1$ to $Rel_2$ as $d_{12}$ and the reverse $d_{21}$. The imbalance in $d_{12}, d_{21}$ can be quantified with $\frac{1}{2}(\frac{d_{12}}{d_{21}} + \frac{d_{21}}{d_{12}})$, which is minimized to 1 when $d_{21} = d_{12}$ and increases as $d_{12}, d_{21}$ are more imbalanced; we call this factor $imb$.

For hypothesis (1), we verified that the vast majority of relation pairs have angles near to $90°$, with the mean and median respectively $83.0°$ and $85.4°$; only $1\%$ of all relation pairs had angles less than $50°$. We observed that relation pairs with angle less than $20°$ were those that can be inferred by transitively applying the pre-determined implication rules. Relation pairs with angles within the range of $[20°, 60°]$ had strong tendencies of semantic relatedness or implication; such tendency drastically weakened past $70°$. Table 4 shows the angle and $imb$ of relations with respect to `/people/person/place_of_birth`, whose trend agrees with our hypotheses. Finally, we note that such an analysis could be possible with TransH as well, since their method too maps $\overrightarrow{t - h}$'s to lines (Fig. 2b).

In all of link Prediction, triple classification, and semantics mining, TransINT's theme of assigning optimal regions to bound entity sets is unified and consistent. Furthermore, the integration of rules into embedding space geometrically coherent with KG embeddings alone. These two qualities were missing in existing works such as TransE, KALE, and SimplE+.

## 7. Related Work

Our work is related to two strands of work. The first strand is Order Embeddings [Vendrov et al., 2015] and their extensions [Vilnis et al., 2018, Athiwaratkun and Wilson, 2018], which are significantly limited in that only unary relations and their hierarchies can be modeled. While Nickel and Kiela [2017] also approximately embed unary partial ordering, their focus is on achieving reasonably competent result with unsupervised learning of rules in low dimensions, while ours is achieving state-of-the-art in a supervised setting.

The second strand is those that enforce the satisfaction of common sense logical rules for binary and $n$-ary relations in the embedded KG. Wang et al. [2015a] explicitly constraints the resulting embedding to satisfy logical implications and type constraints via linear programming, but it only requires to do so during inference, not learning. On the other hand, Guo et al. [2016], Rocktäschel et al. [2015], Fatemi et al. [2018] induce that embeddings follow a set of logical rules during learning, but their approaches involve soft induction instead of hard constraints, resulting in rather insignificant improvements. Our work combines the advantages of both Wang et al. [2015a] and works that impose rules during learning. Finally, Demeester et al. [2016] models unary relations only and Minervini et al. [2017] transitivity only, whose contributions are fundamentally different from us.

## 8. Conclusion

We presented TransINT, a new KG embedding method such that relation sets are mapped to continuous sets in $\mathbb{R}^d$, inclusion-ordered isomorphically to implication rules. Our method is extremely powerful, outperforming existing state-of-the-art methods on benchmark datasets by significant margins. We further proposed an interpretable criterion for mining semantic similarity and implication rules among sets of entities with TransINT.

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

## Appendix A. Proof For TransINT's Isomorphic Guarantee

Here, we provide the proofs for Main Theorems 1 and 2. We also explain some concepts necessary in explaining the proofs. We put $^*$ next to definitions and theorems we propose/ introduce. Otherwise, we use existing definitions and cite them.

### A.1 Linear Subspace and Projection

We explain in detail elements of $\mathbb{R}^d$ that were intuitively discussed. In this and later sections, we mark all lemmas and definitions that we newly introduce with $^*$; those not marked with $^*$ are accompanied by reference for proof. We denote all $d \times d$ matrices with capital letters (ex) $A$) and vectors with arrows on top (ex) $\vec{b}$).

#### A.1.1 LINEAR SUBSPACE AND RANK

The linear subspace given by $A(x - \vec{b}) = 0$ ($A$ is $d \times d$ matrix and $b \in \mathbb{R}^d$) is the set of $x \in \mathbb{R}^d$ that are solutions to the equation; its rank is the number of constraints $A(x - \vec{b}) = 0$ imposes. For example, in $\mathbb{R}^3$, a hyperplane is a set of $\vec{x} = [\ x_1, x_2, x_3]\ \in \mathbb{R}^3$ such that $ax_1 + bx_2 + cx_3 - d = 0$ for some scalars $a, b, c, d$; because vectors are bound by one equation (or its "$A$" only really contains one effective equation), a hyperplane's rank is 1 (equivalently $rank(A) = 1$). On the other hand, a line in $\mathbb{R}^3$ imposes to 2 constraints, and its rank is 2 (equivalently $rank(A) = 2$).

Consider two linear subspaces $H_1, H_2$, each given by $A_1(\vec{x} - \vec{b_1}) = 0, A_2(\vec{x} - \vec{b_2}) = 0$. Then,

$$(H_1 \subset H_2) \Leftrightarrow (A_1(\vec{x} - \vec{b_1}) = 0 \Rightarrow A_2(\vec{x} - \vec{b_2}) = 0)$$

by definition. In the rest of the paper, denote $H_i$ as the linear subspace given by some $A_i(\vec{x} - \vec{b_i}) = 0$.

#### A.1.2 PROPERTIES OF PROJECTION

**Invariance**   For all $\vec{x}$ on $H$, projecting $\vec{x}$ onto $H$ is still $\vec{x}$; the converse is also true.
**Lemma 1** $P\vec{x} = \vec{x} \Leftrightarrow \vec{x} \in H$ [Strang].

**Orthogonality**   Projection decomposes any vector $\vec{x}$ to two orthogonal components - $P\vec{x}$ and $(I - P)\vec{x}$. Thus, for any projection matrix $P$, $I - P$ is also a projection matrix that is orthogonal to $P$ (i.e. $P(I - P) = 0$) [Strang].

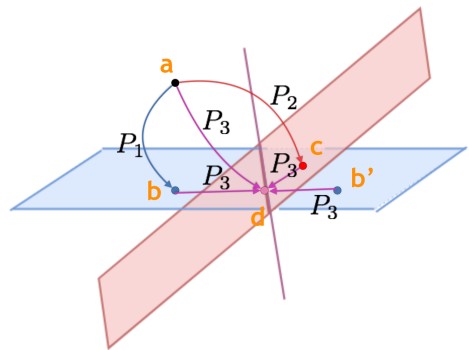

**Figure 5:** Projection matrices of subspaces that include each other.

**Lemma 2** Let $P$ be a projection matrix. Then $I - P$ is also a projection matrix such that $P(I - P) = 0$ [Strang].

The following lemma also follows.

**Lemma 3** $||P\vec{x}|| \leq ||P\vec{x} + (I - P)\vec{x}|| = ||\vec{x}||$ [Strang].

**Projection onto an included space**  If one subspace $H_1$ includes $H_2$, the order of projecting a point onto them does not matter. For example, in Figure 3, a random point $\vec{a}$ in $R^3$ can be first projected onto $H_1$ at $\vec{b}$, and then onto $H_3$ at $\vec{d}$. On the other hand, it can be first projected onto $H_3$ at $\vec{d}$, and then onto $H_1$ at still $\vec{d}$. Thus, the order of applying projections onto spaces that includes one another does not matter.

If we generalize, we obtain the following two lemmas (Figure 5):

**Lemma 4*** Every two subspaces $H_1 \subset H_2$ if and only if $P_1 P_2 = P_2 P_1 = P_1$.

**proof)** By Lemma 1, if $H_1 \subset H_2$, then $P_2 \vec{x} = \vec{x}$ $\forall \vec{x} \in H_1$. On the other hand, if $H_1 \not\subset H_2$, then there is some $\vec{x} \in H_1, \vec{x} \notin H_2$ such that $P_2 \vec{x} \neq \vec{x}$. Thus,

$$H_1 \subset H_2 \Leftrightarrow \forall \vec{x} \in H_1, \quad P_2 \vec{x} = \vec{x}$$
$$\Leftrightarrow \forall \vec{y}, \quad P_2(P_1 \vec{y}) = P_1 \vec{y} \Leftrightarrow P_2 P_1 = P_1.$$

Because projection matrices are symmetric [Strang],

$$P_2 P_1 = P_1 = P_1{}^T = P_1{}^T P_2{}^T = P_1 P_2 . \blacksquare$$

**Lemma 5*** For two subspaces $H_1, H_2$ and vector $\vec{k} \in H_2$,

$$H_1 \subset H_2 \Leftrightarrow Sol(P_2, \vec{k}) \subset Sol(P_1, P_1 \vec{k}).$$

**proof)** $Sol(P_2, \vec{k}) \subset Sol(P_1, P_1 \vec{k})$ is equivlaent to $\forall \vec{x} \in \mathbb{R}^d, P_2 \vec{x} = \vec{k} \Rightarrow P_1 \vec{x} = P_1 \vec{k}$.

By Lemma 4, if $H_1 \subset H_2 \Leftrightarrow P_1 P_2 = P1$. Since $\vec{k} \in P_2$, $P_2 \vec{x} = \vec{k} \Leftrightarrow P_2(x - \vec{k}) = \vec{0} \Leftrightarrow P_1(P_2 \vec{x} - \vec{k}) = \vec{0} \Leftrightarrow P_1 P_2 \vec{x} = P_1 \vec{k} \Leftrightarrow P_1 \vec{x} = P_1 \vec{k} . \blacksquare$

**Partial ordering**  If two subspaces strictly include one another, projection is uniquely defined from lower rank subspace to higher rank subspace, but not the other way around. For example, in Figure 3, a point $\vec{a}$ in $R^3$ (rank 0) is always projected onto $H_1$ (rank 1) at point $\vec{b}$. Similarly, point $\vec{b}$ on $H_1$ (rank 1) is always projected onto similarly, onto $H_3$ (order 2) at point $d$. However, "inverse projection" from $H_3$ to $H_1$ is not defined, because

not only $\vec{b}$ but other points on $H_1$ (such as $\vec{b'}$) project to $H_3$ at point $\vec{d}$; these points belong to $Sol(P_3, \vec{d})$. In other words, $Sol(P_1, \vec{b}) \subset Sol(P_3, \vec{d})$. This is the key intuition for isomorphism , which we prove in the next chapter.

## A.2 Proof for Isomorphism

Now, we prove that TransINT's two constraints (section 2.3) guarantee isomorphic ordering in the embedding space.

Two posets are isomorphic if their sizes are the same and there exists an order-preseving mapping between them. Thus, any two posets $(\{A_i\}_n, \subset)$, $(\{B_i\}_n, \subset)$ are isomorphic if $|\{A_i\}_n| = |\{B_i\}_n|$ and

$$\forall i, j \quad A_i \subset A_j \Leftrightarrow B_i \subset B_j$$

***Main Theorem 1** (Isomorphism):* Let $\{(H_i, \vec{r_i})\}_n$ be the (subspace, vector) embeddings assigned to relations $\{\mathbf{R_i}\}_n$ by the *Intersection Constraint* and *the Projection Constraint*; $P_i$ the projection matrix of $H_i$. Then, $(\{Sol(P_i, \vec{r_i})\}_n, \subset)$ is isomorphic to $(\{\mathbf{R_i}\}_n, \subset)$.
***proof )*** Since each $Sol(P_i, \vec{r_i})$ is distinct and each $\mathbf{R_i}$ is assigned exactly one $Sol(P_i, \vec{r_i})$, $|\{Sol(P_i, \vec{r_i})\}_n| = |\{I_i\}_n|.$①

Now, let's show

$$\forall i, j, \quad R_i \subset R_j \Leftrightarrow Sol(P_i, \vec{r_i}) \subset Sol(P_j, \vec{r_j}).$$

Because the $\forall i, j$, intersection and projection constraints are true iff $\quad R_i \subset R_j$, enough to show that the two constraints hold iff $Sol(P_i, \vec{r_i}) \subset Sol(P_j, \vec{r_j})$.

First, let's show $\mathbf{R_i} \subset \mathbf{R_i} \Rightarrow Sol(P_i, \vec{r_i}) \subset Sol(P_j, \vec{r_j})$. From the *Intersection Constraint*, $\mathbf{R_i} \subset \mathbf{R_i} \Rightarrow H_j \subset H_i$. By Lemma 5, $Sol(P_i, \vec{r_i}) \subset Sol(P_j, P_j\vec{r_i})$. From the *Projection Constraint*, $\vec{r_j} = P_j\vec{r_i}$. Thus, $Sol(P_i, \vec{r_i}) \subset Sol(P_j, P_j\vec{r_i}) = Sol(P_j, \vec{r_j}). \cdots\cdots$ ②

Now, let's show the converse; enough to show that if $Sol(P_i, \vec{r_i}) \subset Sol(P_j, \vec{r_j})$, then the intersection and projection constraints hold true.

$$Sol(P_i, \vec{r_i}) \subset Sol(P_j, \vec{r_j})$$
$$\Leftrightarrow \forall \vec{x}, \quad P_i\vec{x} = \vec{r_i} \Rightarrow P_j\vec{x} = \vec{r_j})$$

If $P_i\vec{x} = \vec{r_i}$,

$$\forall \vec{x}, \quad P_j P_i \vec{x} = P_j \vec{r_i}$$
$$\forall \vec{x}, \quad P_j \vec{x} = \vec{r_j}$$

both have to be true. For any $\vec{x} \in H_i$, or equivalently, if $\vec{x} = P_i\vec{y}$ for some $\vec{y}$, then the second equation becomes $\forall \vec{y}, \quad P_j P_i \vec{y} = \vec{r_j}$, which can be only compatible with the first equation if $\vec{r_j} = P_j\vec{r_i}$, since any vector's projection onto a subspace is unique. (Projection Constraint)

Now that we know $\vec{r_j} = P_j\vec{r_i}$, by Lemma 5, $H_i \subset H_j$ (intersection constraint). $\cdots$ ③ From ①, ②, ③, the two posets are isomorphic.∎

In actual implementation and training, TransINT requires something less strict than $P_i(\overrightarrow{t - h}) = \vec{r_i}$:

$$P_i(\overrightarrow{t - h}) - \vec{r_i} \approx \vec{0} \equiv ||P_i(\overrightarrow{t - h} - \vec{r_i})||_2 < \epsilon,$$

for some non-negative and small $\epsilon$. This bounds $\overrightarrow{t} - \overrightarrow{h} - \overrightarrow{r_i}$ to regions with thickness $2\epsilon$, centered around $Sol(P_i, \overrightarrow{r_i})$ (Figure 4). We prove that isomorphism still holds with this weaker requirement.

**Definition**\* $(Sol_\epsilon(P, k))$ : Given a projection matrix $P$, we call the solution space of $||P\overrightarrow{x} - \overrightarrow{k}||_2 < \epsilon$ as $\mathbf{Sol_\epsilon(P, \overrightarrow{k})}$.

**Main Theorem 2** (Margin-aware Isomorphism): For all non-negative scalars $\epsilon$, $(\{Sol_\epsilon(P_i, \overrightarrow{r_i})\}_n, \subset)$ is isomorphic to $(\{\mathbf{R_i}\}_n, \subset)$.

**proof**) Enough to show that $(\{Sol_\epsilon(P_i, \overrightarrow{r_i})\}_n, \subset)$ and $(\{Sol(P_i, \overrightarrow{r_i})\}_n, \subset)$ are isomorphic for all $\epsilon$.

First, let's show

$$Sol(P_i, \overrightarrow{r_i}) \subset Sol(P_j, \overrightarrow{r_j}) \Rightarrow Sol_\epsilon(P_i, \overrightarrow{r_i}) \subset Sol_\epsilon(P_j, \overrightarrow{r_j}).$$

By Main Theorem 1 and Lemma 4,

$$Sol(P_i, \overrightarrow{r_i}) \subset Sol(P_j, \overrightarrow{r_j}) \Leftrightarrow \overrightarrow{r_j} = P_j\overrightarrow{r_i}, P_j = P_jP_i.$$

Thus, for all vector $\overrightarrow{b}$,

$$P_i(x - \overrightarrow{r_i}) = \overrightarrow{b}$$
$$\Leftrightarrow P_jP_i(\overrightarrow{x} - \overrightarrow{r_i}) = P_j\overrightarrow{b}$$
$$\Leftrightarrow P_j(\overrightarrow{x} - \overrightarrow{r_i}) = P_j\overrightarrow{b} \,(\because \text{Lemma 4})$$
$$\Leftrightarrow P_j(\overrightarrow{x} - \overrightarrow{r_j}) = P_j\overrightarrow{b} \,(\because P_j\overrightarrow{r_j} = \overrightarrow{r_j} = P_j\overrightarrow{r_i})$$

Thus, if $||P_i(\overrightarrow{x} - \overrightarrow{r_i})|| < \epsilon$, then $||P_j(\overrightarrow{x} - \overrightarrow{r_j})|| = ||P_j(P_i(\overrightarrow{x} - \overrightarrow{r_i}))|| < ||P_j(P_i(\overrightarrow{x} - \overrightarrow{r_i})) + (I - P)(P_i(\overrightarrow{x} - \overrightarrow{r_i}))|| = ||P_i(\overrightarrow{x} - \overrightarrow{r_i})|| < \epsilon. \cdots ①$

Now, let's show the converse. Assume $||P_i(\overrightarrow{x} - \overrightarrow{r_i})|| < \epsilon$ for some $i$. Then,

$$||P_j(\overrightarrow{x} - \overrightarrow{r_j})|| = ||P_j(\overrightarrow{x} - \overrightarrow{r_i}) + P_j(\overrightarrow{r_i} - \overrightarrow{r_j})||$$
$$= ||P_j(P_i(\overrightarrow{x} - \overrightarrow{r_i}) + (I - P_i)(\overrightarrow{x} - \overrightarrow{r_i})) + P_j(\overrightarrow{r_i} - \overrightarrow{r_j})||$$
$$= ||P_jP_i(\overrightarrow{x} - \overrightarrow{r_i}) + P_j(I - P_i)(\overrightarrow{x} - \overrightarrow{r_i}) + P_j(\overrightarrow{r_i} - \overrightarrow{r_j})||$$
$$\leq ||P_jP_i(\overrightarrow{x} - \overrightarrow{r_i})|| + ||P_j(I - P_i)(\overrightarrow{x} - \overrightarrow{r_i})|| + ||P_j(\overrightarrow{r_i} - \overrightarrow{r_j})||.$$

$||P_i(\overrightarrow{x} - \overrightarrow{r_i})|| < \epsilon$ bounds $||P_jP_i(\overrightarrow{x} - \overrightarrow{r_i})||$ to at most epsilon. However, because $P, (I - P)$ are orthogonal(Lemma 3) it tells nothing of $||(I - P_i)(\overrightarrow{x} - \overrightarrow{r_i})|| < \epsilon$, and the second term is unbounded.(Figure 5) The third term $||P_j(\overrightarrow{r_i} - \overrightarrow{r_j})||$ is unbounded as well, since $\overrightarrow{r_j}$ can be anything.

Thus, for $||P_i(\overrightarrow{x} - \overrightarrow{r_i})|| < \epsilon$ to bound $||P_j(\overrightarrow{x} - \overrightarrow{r_j})||$ at all for all $\overrightarrow{x}$,

$$P_j(I - P_i) = 0, P_j(\overrightarrow{r_i} - \overrightarrow{r_j}) = 0$$

need to hold. By Lemma 4 and 5,

$$P_j = P_jP_i \Leftrightarrow H_j \subset H_i$$
$$\Leftrightarrow Sol(P_i, \overrightarrow{r_i}) \subset Sol(P_j, P_j\overrightarrow{r_i}) = Sol(P_j, \overrightarrow{r_j}) \cdot\cdot②$$

$|\{Sol_\epsilon(P_i, \overrightarrow{r_i})\}_n| = |\{Sol(P_i, \overrightarrow{r_i})\}_n|$ holds obviously; each $Sol(P_i, \overrightarrow{r_i})$ has a distinct $Sol_\epsilon(P_i, \overrightarrow{r_i})$ and each $Sol_\epsilon(P_i, \overrightarrow{r_i})$ also has a distinct "center" $(Sol(P_i, \overrightarrow{r_i})) \cdot\cdot③$

From ①, ②, ③, the two sets are isomorphic. ∎

## Appendix B. Definition of "Head", "Parent", "Child" relations (section 4.2)

**Definition**[*] *(Parent/ Child Relations):* For two relations $r_1, r_2$, if $r_1 \Rightarrow r_2$ and there is no $r_3$ such that $r_1 \Rightarrow r_2 \Rightarrow r_2$, then $r_1$ is parent of $r_2$; $r_2$ is child of $r_1$. For example, in Figure 1c of the submitted paper, *is_family_of* is *is_parent_of*'s parent.

**Definition**[*] *(head):* A relation $r$ that has no parent is a head relation.

For example, in Figure 1c of the submitted paper, *is_extended_family_of* is a head relation and its *max_len* is 4.

## Appendix C. Explanation on NELL Sport/ Location (section 5)

Here are the rules contained in NELL Sport/ Location, copied from [Wang et al., 2015a] and [Fatemi et al., 2018].

**Table 5:** Relations and Rules in Sport and Location datasets.

| | Relations | Rules |
|---|---|---|
| **Sport** | AthleteLedSportsTeam
AthletePlaysForTeam
CoachesTeam
OrganizationHiredPerson
PersonBelongsToOrganization | $(x, AtheleLedSportsTeam, y) \rightarrow (x, AthletePlaysForTeam, y)$
$(x, AthletePlaysForTeam, y) \rightarrow (x, PersonBelongsToOrganization, y)$
$(x, CoachesTeam, y) \rightarrow (x, PersonBelongsToOrganization, y)$
$(x, OrganizationHiredPerson, y) \rightarrow (y, PersonBelongsToOrganization, x)$
$(x, PersonBelongsToOrganization, y) \rightarrow (y, OrganizationHiredPerson, x)$ |
| **Location** | CapitalCityOfCountry
CityLocatedInCountry
CityLocatedInState
StateHasCapital
StateLocatedInCountry | $(x, CapitalCityOfCountry, y) \rightarrow (x, CityLocatedInCountry, y)$
$(x, StateHasCapital, y) \rightarrow (y, CityLocatedInState, x)$ |