# OpenReview forum: "TransINT: Embedding Implication Rules in Knowledge Graphs with Isomorphic Intersections of Linear Subspaces"
_AKBC.ws/2020/Conference — AKBC 2020_

### Official Review · AnonReviewer1 · 2020-03-27
**Good motivation but evaluation on more datasets would be more convincing**

**Rating:** 6
**Confidence:** 3

**Review:**

This work proposes a new knowledge graph embedding method in the Trans- family, that ensures the implication ordering of relations in the embedding space. The evaluation of the proposed model is done on a single dataset - FB122, in which it outperforms previous Trans models.

Pros:
- The proposed method is well motivated and described in detail.
- In the current evaluation, the proposed method outperforms the previous model in the same family with a large margin.
- The resulted embeddings seem to encode some semantic relatedness which can be considered interpretable. Here the claim of verifying the hypothesis could be backed with more examples than the few in table 3 - by placing them in the appendix rather than “our code repository”.
- It seems that the code will be published, although this statement is not explicitly made.

Cons:
- The evaluation is done only on one dataset, while related work evaluates their methods on other datasets such as WN18 and NELL. Why aren’t other standard datasets considered - such as WN18? Are there limitations of the model that are not discussed or what?
- More details on the implementation/evaluation would be nice:
-- In terms of optimization, only the standard loss is discussed. In section  “Automatic Grounding of Positive Triples” - how exactly the implication constraints applied during training?
-- What does mean “we create 100 mini-batches of the training set”.What is the size of FB122 and why exactly 100 mini-batches?
-- What framework is used for the implementation?
-- What strategy and how many configurations are used for finding the optimal hyper-parameters?

Minor:
The distance between Tables 2 and 3 does not seem natural and is making the reading hard. The column in Table 3 is called “imb”, but there is enough space to write the whole name.

---

> ### Author Response · Authors · 2020-04-17
> **Thank you for your feedback. Here are our responses.**
>
> Thank you very much for giving insightful feedback on our work. We have updated the paper with new experiments (significantly outperforming SimplE+ on NELL Sport/ Location) and better readability - please check the common response in the comment above and our updated paper! We have also fixed the \vspace’s and unnatural distances between tables. Here are our responses to your questions.
>
> 1. In terms of optimization, only the standard loss is discussed. In section  “Automatic Grounding of Positive Triples” - how exactly the implication constraints applied during training?
>
> In Section 4.2, we had meant that our parameter sharing initialization automatically applies implication constraints throughout all stages of training, which enables the use of the standard loss and no special treatment during training. We agree with you that this point was not explained well enough, and improved the clarity of Section 4.2.
>
> 2. What does mean “we create 100 mini-batches of the training set”.What is the size of FB122 and why exactly 100 mini-batches?
>
> FB122 has 9738 entities and 122 relations. Its training/ validation/ test set respectively have  91638, 9595, and 11243 (head, relation, tail) entries. In creating 100 mini-batches, we followed the protocol of KALE, which experimented on the same dataset.
>
>  3. What framework is used for the implementation?
>
> We used Pytorch. We now mention this in the beginning of the “Experiments” section.
>
> 4. What strategy and how many configurations are used for finding the optimal hyper-parameters?
>
> We made this information more clear in Section 5.1.
>
> [1]: Guo, S.; Wang, Q.; Wang, L.; Wang, B.; and Guo, L. 2016. Jointly embedding knowledge graphs and logical rules. In EMNLP, 192–202.

---

### Official Review · AnonReviewer2 · 2020-03-28
**Interesting angle but not very clear and not enough evidence**

**Rating:** 4
**Confidence:** 3

**Review:**

The paper proposes the idea of a new knowledge graph embedding technique that builds on top of TransH and incorporate implication ordering. Results show that it outperforms the previous state-of-the-art method on link prediction and triple classification tasks on FB122. The idea and the new angle of looking at TransH are interesting, but the paper needs lots of revision in terms of clarity and formatting. More experiments could also be included to better show strength.

Pros:
- The use of the idea that relations can be viewed as sets of pairs of entities is intriguing and different from most previous KG embedding approaches
- The new angle provided for TransH embedding is also worth learning

Cons:
- Readability of this paper is low, due to intentionally adding lots of "-vspace" (or equivalent) to fit in the page limit. While I understand the limit is mandatory, lots of the sentences could be rephrased and figures could be rearranged instead of removing white space between lines/formulas/figures/tables, which will make it very hard for readers to follow.
- The paper is not very clearly written. Some sentences, such as the last sentence in Intro paragraph 1 and the second sentence from Intro paragraph 2, are not clear to me even after finish reading the paper.  Also, some figures are not very clearly illustrated and the same goes for the captions, especially figures 2 and 3. Typos and unclosed parenthesis also exist.
- Experiments, although shows promising results, may not be comprehensive enough to show the strength of the proposed model. It only tests on a subset of the KG embedding tasks and compares TransINT with a few models.

In summary, this paper needs to be improved in terms of clarity, readability, and the strength of experiments and is currently not ready for publication at AKBC.

---

> ### Author Response · Authors · 2020-04-17
> **Thank you for your feedback. We updated the paper with more experiments.**
>
> Thank you very much for giving insightful feedback on our work. We have updated the paper with new experiments (significantly outperforming SimplE+ on NELL Sport/ Location) and better readability - please check the common response in the comment above and our updated paper!

---

### Official Review · AnonReviewer3 · 2020-03-29
**Good focused contribution, writing can be significantly improved**

**Rating:** 6
**Confidence:** 3

**Review:**

This work describes a new approach to learn KG embeddings by preserving the "implication ordering" among relations in the embedding space. The paper provides a cute new interpretation of TransH and then uses it to extend TransH to learn entity and relation representations. The crucial novelty of the approach is to map relations into linear subspaces whose parameters are tied using the implication ordering of relations.

The paper has clear value and the experiments are sound (although only on one dataset). I find it hard to evaluate its novelty at the moment because there is a clear lack in discussion of prior work in this paper, but I like it overall.

Question for authors: Is this the first method that uses the implication ordering of relations for KG representation learning?
Why is the method only compared to only trans based rule integration methods? Why are the other methods not discussed and compared against?

I think the related work section of this paper needs to be significantly expanded and the method should be contrasted with more works both in the experiments as well in the related work discussion.

Also, why are all the evaluations only one the small FB122 dataset. Why are larger datasets not considered?

Where does the ordering of relations come from? I could not find this in the paper.

The paper is very compressed - the spacing between many lines is very low. I would recommend cutting down on sections 1, 2.1 and 2.2 and expanding on the rest.

What are the number of parameters in this model. How does it compare to the original TransH method? How does the performance change with less/more data.

page 6: what is c_i? why is this needed?

The paper describes a hard parameter sharing scheme. Can we compare this to softer constraints or other variations of parameter sharing as an ablation study?

I find the claim of the "angles between the continuous sets as interpretable" suspicious. How is this more interpretable than other methods which use vector operations or distances for the same? Particularly, how would we quantify this claim?

Typos:
page 5:
 r_i instead of r_1
scalars instead of scalar
page 7:
ignores instead of ignore
page 9:
the tables should have TransINT^G and TransINT^NG
page 10:
similarity

---

> ### Author Response · Authors · 2020-04-17
> **Thank you for your feedback. Here are our responses.**
>
> Thank you very much for giving insightful feedback on our work. Here are our responses to your questions. We have updated the paper with new experiments (significantly outperforming SimplE+ on NELL Sport/ Location) and better readability - please check the common response in the comment above and our updated paper! We have also fixed the typos/ mistakes that you (thankfully) have caught and mentioned.
>
> 1. Is this the first method that uses the implication ordering of relations for KG representation learning?
>
> Answer: No, works such as [1,2,3,4,5] have previously used implication ordering of relation for KG representation learning. We improved the “Related Work” section, by including more previous works that have used implication ordering, stating our method’s over them, and enhancing the readability.
>
> 2. Where does the ordering of relations come from? I could not find this in the paper.
>
> TransINT, like other rule-integrating KG embedding methods [1, 2],  assumes pre-defined rules (such as is_father_of -> is_parent_of) to be given together with knowledge graph training data. We have made this more clear by adding in the abstract and the introduction, by adding “given implication rules” before explaining TransINT’s mechanism.
>
> 3. What are the number of parameters in this model. How does it compare to the original TransH method? How does the performance change with less/more data.
>
> TransINT always has the same number of parameters as TransH, because both TransINT and TransH assign 2 d-dimensional vectors to every relation, and one d-dimensional vector to every entity. We explained this with better readability in section 4.1.
>
> Because we tested our models against benchmark datasets, we did not do an ablation study by changing the amount of training data. However, NELL sports/ locations (a dataset that we newly included results on) is a much smaller dataset than FB 122; TransINT had no problem learning with this smaller dataset.
>
> 4. The paper describes a hard parameter sharing scheme. Can we compare this to softer constraints or other variations of parameter sharing as an ablation study?
>
> While the question is valid and insightful, I do not think ablation study with softer constraints is possible on our method. However, the methods we compare with (KALE [1] and SimplE+ [2]), are both methods with softer constraints, both of which we outperform with significant gaps in our experiments. Especially, SimplE+ is based on SimplE, a very new and powerful knowledge graph embedding method, but still underperforms compared to our method. We think this is one piece of evidence that hard constraints can be very powerful.
>
> 5. I find the claim of the "angles between the continuous sets as interpretable" suspicious. How is this more interpretable than other methods which use vector operations or distances for the same? Particularly, how would we quantify this claim?
>
> We agree that we cannot quantify or claim that the angles between our linear subspaces is a more powerful/ interpretable measure than other kinds of operations for semantic-relatedness mining used in other methods. However, we still think that our angle-based approach has merits in two senses. First, the angle between any two relations can be efficiently calculated with the inner product of their orthogonal subspaces.  Furthermore, angle is a geometrically intuitive measure.
>
> [1]: Guo, S.; Wang, Q.; Wang, L.; Wang, B.; and Guo, L. 2016. Jointly embedding knowledge graphs and logical rules. In EMNLP, 192–202.
> [2]: Fatemi, Bahare et al. “Improved Knowledge Graph Embedding using Background Taxonomic Information.” AAAI (2019).
> [3]: Ivan Vendrov, Jamie Ryan Kiros, Sanja Fidler, and Raquel Urtasun. Order-embeddings of images and language. CoRR, abs/1511.06361, 2015.
> [4]: Luke Vilnis, Xiang Li, Shikhar Murty, and Andrew McCallum. Probabilistic embedding of knowledge graphs with box lattice measures. arXiv preprint arXiv:1805.06627, 2018.
> [5]: Tim Rockt ̈aschel, Sameer Singh, and Sebastian Riedel. Injecting logical background knowledge into embeddings for relation extraction. In Proceedings of the 2015 Con- ference of the North American Chapter of the Association for Computational Linguis- tics: Human Language Technologies, pages 1119–1129, Denver, Colorado, May–June 2015. Association for Computational Linguistics. doi: 10.3115/v1/N15-1118.

---

### Author Response · Authors · 2020-04-17
**Thank you for your feedback. We revised our draft.**

Dear reviewers,

Thank you very much for providing detailed and insightful feedback on our work. We have updated the paper with new results on the NELL Sport/ Location dataset [1] with comparison to SimplE+ [2], a state-of-the-art knowledge graph embedding method that integrates implication rules. Our method TransINT again outperformed SimplE+ with a great margin on all metrics. Furthermore, we have also made the paper more readable by improving clarity of sentences and refraining from using “-vspace”’s.

We will be very thankful if you could check a new version of the paper, and see if the points above are appropriately addressed. Again, we thank you for your detailed feedback.

[1]: Quan Wang, Bin Wang, and Li Guo. 2015. Knowledge base completion using embeddings and rules. In Proceedings of the 24th International Conference on Artificial Intelligence (IJCAI’15). AAAI Press, 1859–1865.
[2]: Fatemi, Bahare et al. “Improved Knowledge Graph Embedding using Background Taxonomic Information.” AAAI (2019).

---

> ### Comment · AnonReviewer1 · 2020-04-29
> **Updates highlighting**
>
> Thank you for the updates and for addressing mine and other reviewer's questions. However, it would have been nice to highlight the changes using a blue font or some other way because it is hard to see them.

---

### Decision · Program_Chairs · 2020-04-30

**Decision:**

Accept

**Comment:**

This work proposes a new knowledge graph embedding method in the Trans- family, that ensures the implication ordering of relations in the embedding space. The proposed idea on viewing relations as sets of pairs of entities is interesting and provides new perspective as compared to previous KG embedding approaches. The technical content is well explained and justified. There are concerns from the reviewers on experiments and writing. The author has revised the draft to incorporate the review comments.